# Feasibility of Secondary Follicle Isolation, Culture and Achievement of In-Vitro Oocyte Maturation from Superovulated Ovaries: An Experimental Proof-of-Concept Study Using Mice

**DOI:** 10.3390/jcm10132757

**Published:** 2021-06-23

**Authors:** Xia Hao, Amandine Anastácio, Kenny A. Rodriguez-Wallberg

**Affiliations:** 1Department of Oncology and Pathology, Karolinska Institutet, Solna, SE-171 76 Stockholm, Sweden; xia.hao@ki.se (X.H.); anastacio.amandine@gmail.com (A.A.); 2Laboratory of Translational Fertility Preservation, BioClinicum, SE-171 64 Stockholm, Sweden; 3Department of Reproductive Medicine, Division of Gynecology and Reproduction, Karolinska University Hospital, Novumhuset Plan 4, SE-141 86 Stockholm, Sweden

**Keywords:** female fertility preservation, superovulation, follicle isolation, secondary follicle culture, in-vitro maturation, ovarian tissue

## Abstract

Fertility preservation through ovarian stimulation, aiming at cryopreserving mature oocytes or embryos, is sometimes unsuccessful. This clinical situation deserves novel approaches to overcome infertility following cancer treatment in patients facing highly gonadotoxic treatment. In this controlled experimental study, we investigated the feasibility of in-vitro culturing secondary follicles isolated from superovulated ovaries of mice recently treated with gonadotropins. The follicle yields of superovulated ovaries were 45.9% less than in unstimulated controls. Follicles from superovulated ovaries showed faster growth pace during the initial 7 days of culture and secreted more 17β-estradiol by the end of culture *vs* controls. Parameters reflecting the outcome of follicular development and oocyte maturation competence in vitro were similar between superovulated and control groups, with a similar follicle size at the end of culture and around 70% survival. Nearly half of cultured follicles met the criteria for in-vitro maturation in both groups and approximately 60% of those achieved a mature MII oocyte, similarly in both groups. Over 60% of obtained MII oocytes displayed normal-looking spindle and chromosome configurations, without significant differences between the groups. Using a validated follicle culture system, we demonstrated the feasibility of secondary follicle isolation, in-vitro culture and oocyte maturation with normal spindle and chromosome configurations obtained from superovulated mice ovaries.

## 1. Introduction

Effective cancer screening, early diagnosis and advanced treatment have improved cancer patients’ long-term survival rate [1,2]. However, as an off-target effect from radiotherapy and chemotherapy, the development of infertility is negatively impacting cancer survivors’ life quality [3]. In adult female patients, cryopreservation of mature oocytes and embryos are well-established fertility preservative methods [4,5,6,7]. In this context, an increasing population of women undergo controlled ovarian stimulation (COS) treatment with gonadotropins worldwide, aiming at the retrieval of multiple mature oocytes in a single cycle. The clinical pregnancy rate of using cryopreserved mature oocytes that are fertilized and result in an embryo transfer is around 40% [8,9]. One cycle of COS treatment typically requires two weeks to be completed. In some cases, a poor ovarian response to hormonal stimulation may be evident, and the women do not gain any frozen oocytes or embryos from the COS treatment. As patients with cancer should not delay initiation of oncologic treatment, there is a lack of time to plan for additional COS cycles. Alternative fertility preservative options need to be developed for those cases.

Cryopreservation of ovarian tissue is an alternative method for female fertility preservation [10]. The method requires surgery, usually a minimally invasive laparoscopy that can be planned shortly, and it does not require hormonal stimulation. The success rate of ovarian tissue transplantation is difficult to accurately calculate, as many cases may still remain unreported, however the data provided from a Spanish fertility preservation program reported a 46.7% spontaneous pregnancy rate [11], while a meta-analysis reported a live birth and ongoing pregnancy rate of 37.7% [12]. The methods for ovarian tissue cryopreservation provide an effective preservation of small primordial follicles representing the ovarian pool, while the growing follicles including those at the preantral stage do not usually survive after thaw [13,14]. Collection of cumulus-oocyte complexes (COCs) and denuded oocytes followed by in-vitro maturation (IVM) to improve fertility preservation options during ovarian tissue cryopreservation have been reported [15,16,17]. The use of secondary follicles reported in this study as a resource of mature oocytes has not been previously reported.

The study was designed as a proof-of-concept aiming at developing a method to rescue follicles after a failed COS attempt. In such clinical situations, the cryopreservation of ovarian tissue is not recommended, as the ovarian morphology is highly affected, and thus the cryopreservation of the ovarian cortex is unlikely to be successful according to the currently standardized methods [18]. In this study, the feasibility of secondary follicle isolation and in-vitro growth up to mature oocytes from superovulated mice ovaries was investigated. The culture of secondary follicles from unstimulated ovaries and achievement of mature oocytes has been previously described and used over the years [19,20,21]. In our controlled experimental study, C57BL/6J mice undergoing in-vitro fertilization (IVF) were used. The mice strain has been previously validated and is widely used in basic research on fertility preservation [22,23,24,25,26]. The age of the mouse was 4–5 weeks to mimic adult women of a young reproductive age. A control group of mice ovaries without hormonal stimulation was used to compare follicle development during in-vitro culture and maturation competence after IVM. Several in-vitro culture outcomes including follicle attachment to culture dish, follicular sizes and growth curve, antrum-like cavity formation, 12 days’ survival rate and secreted 17β-estradiol were evaluated to assess follicular growth. In addition, selected follicles were matured in vitro and oocyte maturation outcomes were evaluated as well as the spindle and chromosome configurations of obtained mature oocytes.

## 2. Materials and Methods

### 2.1. Chemicals

All chemicals used in this study were purchased from Sigma–Aldrich^®^ (St. Louis, MO, USA) or Gibco, Thermofischer Scientific^®^ (Paisley, UK), unless otherwise stated.

### 2.2. Animals and Grouping

For this study, 28 4–5 week old C57BL/6J female mice were used. The mice were included in two groups, either undergoing treatment with gonadotropins and superovulation for IVF at the experimental research facility PKL5, Novum, Karolinska Institutet, Huddinge, (n = 19), or used as a control group without stimulation (n = 9). The superovulation was induced with an intraperitoneal injection of pregnant mare’s serum gonadotrophin (5 IU, Folligon^®^, MSD animal health, Brussel, Belgium) followed by an injection of human chorionic gonadotrophin (hCG, 5 IU, Chorulon^®^, MSD animal health, Boxmeer, Holland) 50 h later. Mice were sacrificed 13.5 h after hCG stimulus for collection of the ovaries and the cumulus-oocyte complexes (COCs). All procedures were performed in three independent trails.

All the experimental procedures described were carried out in agreement with ethics permits (03711-2020 and 1372-17) accorded by Karolinska Institutet’s Ethics Committee for research using experimental laboratory animals.

### 2.3. Follicle Isolation and Secondary Follicle Culture

Ovaries were collected in dissection medium which was Leibovitz-15 medium supplemented with 10% fetal bovine serum (FBS), 100 IU/mL of penicillin and 100 µg/mL of streptomycin, and kept at 4 °C, if not used immediately, for a maximum of 6 h.

Secondary follicles were mechanically isolated in a dissection medium using micro-fine U-100 insulin syringes (0.3 mL, BD Medical, Le Pont-de-Claix Cedex, France) under stereo-microscope (Nikon^®^, Tokyo, Japan). Intact follicles with a diameter of 100–130 µm, with two or more granulosa layers, with few attached theca cells and with a visible, round and central oocyte were selected for in-vitro culture [19,20,21]. The selected follicles were cultured individually in culture dishes (TC-Schale 60, Standard, Nümbrecht, Germany) containing 10 × 10 µL culture medium droplets, covered with 5 mL of mineral oil. The culture medium used was α-minimal essential medium (α-MEM) GlutaMAX enriched with 5% FBS, 5 µg/mL insulin, 10 µg/mL transferrin, 100 mIU/mL recombinant follicle stimulating hormone (GONAL-F, Merck Europe B.V., Amsterdam, Holland). The follicles were cultured for 12 days in a humidified incubator at 37 °C and with 5% CO_2_. The day the follicles were selected for culture was designated as Day 0 and the last day as Day 12. On Day 1 of culture, 10 µL of culture medium was added to each droplet. Thereafter, half of the medium was refreshed every other day and the follicles were observed under an inverted microscope (Nikon^®^, Tokyo, Japan) for morphological analysis and measurement. Follicle diameter was assessed using a calibrated ocular micrometer, and two perpendicular measures including the granulosa cell mass without the theca cells of each follicle were registered. The collected medium was diluted in 90 µL of α-MEM with bovine serum albumin (BSA, 40 mg/mL) and stored at −20 °C for further use. On Day 12 of culture, follicles with a size ≥ 200 µm, presenting a clear granulosa cell proliferation and a visible round oocyte were classified as growing follicles.

### 2.4. Oocyte In-Vitro Maturation

Follicles with at least 400 µm of diameter on day 12 were selected for IVM. Oocytes of selected follicles with some surrounding granulosa cells were individually transferred under stereo-microscope to maturation dishes (TC-Schale 60, Standard, Nümbrecht, Germany) containing 10 × 20 µL maturation medium droplets covered with 5 mL of mineral oil, then incubated at 37 °C, with 5% CO_2_. The maturation medium was culture medium supplemented with 1.5 IU/mL recombinant hCG and 5 ng/mL recombinant epidermal growth factor. Maturation dishes were equilibrated in the incubator overnight before use.

Formation of COCs were verified 16–20 h after incubation and oocyte denudation was performed to evaluate the oocyte maturation status. Oocytes with a visible polar body were classified as mature (Metaphase II, MII) or immature (germinal vesicle, GV) if a germinal vesicle was visible. Oocytes with neither polar body nor germinal vesicle were classified as Metaphase I (MI).

### 2.5. Spindle and Chromosome Configuration Analysis

After IVM, all denuded mature MII oocytes in superovulated and control groups were collected and washed in Dulbecco’s Phosphate-Buffered Saline containing 0.1% polyvinyl alcohol (washing buffer). Thereafter, oocytes were fixed with 2% formaldehyde in washing buffer containing 0.2% Triton X-100 for 40 min. After fixation, the oocytes were incubated in blocking buffer (washing buffer supplemented with 1% BSA) overnight at 4 °C. They were blocked for 40 min with a blocking buffer containing 10% FBS. Then, oocytes were incubated with mouse monoclonal anti-α-tubulin antibody (T9026, 1:1000) in blocking buffer for 45 min followed by Alexa Fluor 488-labelled goat anti-mouse IgG H&L (ab150113, Abcam, UK, 1:200) in a blocking buffer for 40 min at 37 °C. Then, oocytes were incubated with 10 μg/mL propidium iodide (81845) for 20 min. At the end, oocytes were mounted with Prolong Diamond Antifade mountant (P36965, Invitrogen, Eugene, Oregon) between a cover slip and a microscope slide. Slides were kept at 4 °C until Confocal imaging.

Labelled tubulin and chromatin were assessed using Nikon Eclipse Ti microscope (Nikon^®^, Tokyo, Japan) equipped with the appropriate filter sets for analyzing Alexa Fluor 488 and propidium iodide with 100 × oil immersion objective. Oocyte images were captured with Andor iXon Ultra and analyzed by NIS-Elements program. Image analyses were performed using ImageJ 1.53c software (National Institutes of Health, Bethesda, MD, USA).

### 2.6. Hormone Assay

Culture medium samples were collected on Day 5, 9 and 12 and used to measure the secretion of 17β-estradiol using commercially available enzyme-linked immunoassay kit, 17β-estradiol (ab108667, Abcam, Germany), following the manufacturers’ protocols. The limit of sensitivity for 17β-estradiol was 8.68 pg/mL. For each estimation time point in each group, the culture medium collected from 5 follicles with similar growth features and from which obtained mature oocytes were pooled together to reach the required volume of sample amount for the essay. The measurements were performed in triplicates.

### 2.7. Statistical Analysis

Statistical analyses were performed using the GraphPad Prism 8.4.3 software (San Diego, CA, USA). Comparisons between the superovulated and the control groups regarding follicular sizes, 17β-estradiol levels on different culture days and the numbers of isolated secondary follicles were compared by t-tests. Comparisons of the proportions of follicle attachments, antrum-like cavity formation, survival and IVM outcomes and percentages of MII oocytes with spindle defects or chromosome misalignment between superovulated and control group were all performed by Chi-square test (two-sided).

## 3. Results

### 3.1. Secondary Follicle Isolation, In-Vitro Development and Maturation

Isolated secondary follicles yield meeting the criteria for in-vitro culture in the superovulated group was nearly half of the yield observed in the control group (*p* < 0.05, Table 1). On Day 1 of culture, 78.7% of follicles in the superovulated group attached to the bottom of the culture dishes and this ratio increased to 93.3% on Day 3, which was significantly higher compared with the attachment ratio found in the control group (42% on Day 1 and 76.5% on Day 3). At the end of culture, similar percentages of follicles had survived in both groups (Table 1). Additionally, antrum-like cavities were observed in 8.9% of the follicles isolated from the superovulated group and 10% of the follicles from the control group.

On Day 12, a similar proportion of follicles in each group fulfilled the criteria to undergo IVM and similar percentages of mature MII oocytes were obtained in superovulated (63.9%) and control (59.6%) groups (*p* = 0.5180) (Table 1).

The growth behavior and trend during culture displayed by growth curves were similar between the groups of follicles obtained from superovulated and control ovaries, Figure 1a. However, follicles in the superovulated group grew significantly larger than those in the control group between Day 1 and Day 7, while since Day 9, follicles in both groups reached similar mean follicular sizes (Table 2).

### 3.2. Secretion of 17β-Estradiol during Culture, Spindle and Chromosome Analysis in Mature Oocytes

Follicle secretion of 17β-estradiol determined in culture media slightly increased and reached a similar level in the first 9 days of culture in superovulated and control groups, thereafter till Day 12, the level of 17β-estradiol markedly increased (by 1733.8 pg/mL) in the superovulated group, whereas the increase was smaller in control group (by 142.7 pg/mL), Figure 1b. On Day 12, the level of 17β-estradiol in the superovulated group was significantly higher than control (*p* < 0.05).

MII oocytes with normal- and abnormal-looking spindle and chromosome structures were observed in both groups, Figure 1c. The normal-looking MII oocytes displayed bipolar barrel-shaped spindles and well-aligned chromosomes on the metaphase equator, whereas abnormal-looking MII oocytes showed spindle defects and/or chromosome misalignment. About 30–40% of MII oocytes presented with chromosome misalignment or spindle defects, and the proportions of abnormal-looking MII oocytes were similar between the superovulated and control groups, Figure 1d.

## 4. Discussion

In this study, we used ovaries of mice recently superovulated to experimentally investigate the feasibility to obtain secondary follicles in this specific condition. The efficacy of this approach was also investigated by comparing the outcomes with those of follicles retrieved from unstimulated control ovaries. Outcomes of follicles isolated from superovulated ovaries undergoing culture and maturation in vitro indicated the feasibility of this approach. Additionally, normal spindle configurations and chromosome structures were found in about 60% of the cases, which was similar and did not differ significantly from those found in follicles from unstimulated control groups. The achieved similar final developmental and maturation outcomes in both groups reflected the efficiency of the method. However, the lower numbers of secondary follicles retrieved from superovulated ovaries meeting the criteria for culture might be a consequence of a successful recruitment of growing follicles using high dose of gonadotropins, and thus fewer secondary follicles were left in these ovaries in comparison with the unstimulated control group.

During culture, follicular growth curves and trends were similar between groups, with slower growth during the first 5 days of culture and faster growth thereafter. Thus, levels of 17β-estradiol were measured firstly on Day 5 and at two additional timepoints thereafter, including the day of the end of culture. The increasing secretion of 17β-estradiol showed a correlation with the increase of follicular sizes, as expected [27,28,29].

It was observed under the culture that follicles from superovulated ovaries seemed to adapt faster to the culture system than control follicles, as indicated by the significantly higher percentages of follicles attached to the culture dish during the initial days of culture. These follicles also grew faster initially, since the start of culture, compared to the control group. A possible explanation is that the gonadotropins previously received in-vivo exerted a continuing effect on the proliferation and function of granulosa cells, reflected also in the higher levels of 17β-estradiol secreted in the superovulated follicle group *vs* controls. Apparently, this effect did not influence the final outcomes of the culture, as similar percentages of cultured follicles formed an antrum-like cavity, survived the whole culture and met the selection criteria to be put into IVM in both groups. The timing and follicle selection for IVM were according to the final follicle size achieved on Day 12 of culture [20,21].

The maturation ability of oocytes following hCG-induced IVM was also preserved in the group of follicles retrieved from superovulated ovaries, and similar percentages of MII oocytes were obtained in both groups. Additionally, similar percentages of MII oocytes with normal spindle structure and chromosome configurations were also found, without significant differences between the groups. Our study provides indications that support the retrieval of preantral secondary follicles for in-vitro culture even after COS. This is proposed as a rescue method, and it should be further investigated aiming at finding novel fertility preservative methods to women who have failed to produce mature oocytes for cryopreservation after cycle(s) of COS treatment. The method proposed herein could offer a resource for obtaining mature oocytes through collecting the ovarian tissue and performing secondary follicles isolation, in-vitro culture and IVM.

Further development of the method hereby described is needed for the establishment of clinically standardized fertility preservative methods. Use of human materials in experimental research will increase the similarity to the clinical setting instead of using mice. There are established methods for conducting human secondary follicle in-vitro culture and oocytes IVM isolated from human ovarian cortical strips [30,31]. Additional investigation of the fertilization competence of mature oocytes obtained in vitro through this approach should be conducted, and importantly, the follow-up of their offspring.

## Figures and Tables

**Figure 1 jcm-10-02757-f001:**
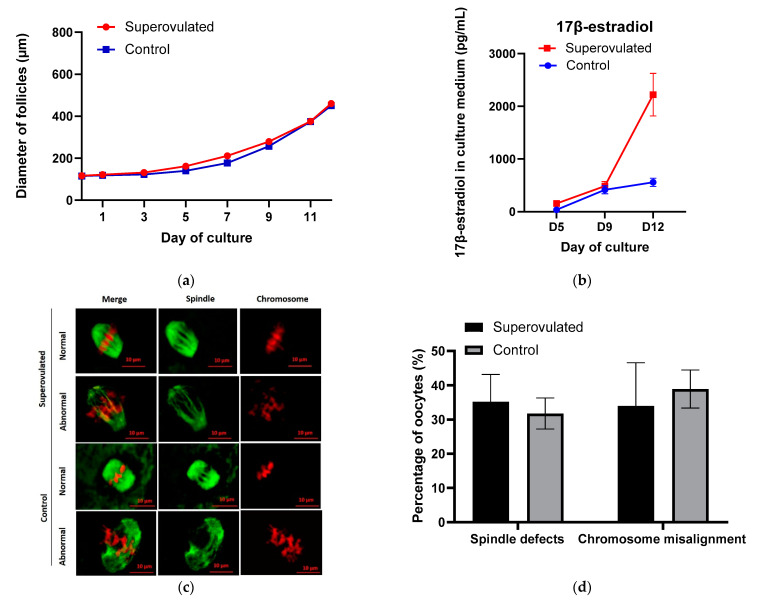
(**a**) Follicular growth curves during in-vitro culture; (**b**) 17β-estradiol secreted by individual follicles in the culture medium; (**c**) Representative images showing a normal or abnormal morphology of spindle and chromosome configurations in MII oocytes from control and superovulated groups after IVM. Spindle fibers (green) were detected by immunofluorescence for a-tubulin, while DNA (red) was stained with propidium iodide. Scale bar, 10 µm; (**d**) Percentages of MII oocytes with spindle defects or chromosome misalignment in control and superovulated groups.

**Table 1 jcm-10-02757-t001:** Comparison of parameters reflecting follicular growth and oocyte maturation competence (* *p* < 0.05, ** *p* < 0.01, *** *p* < 0.001), MI: metaphase I, MII: metaphase II, GV: germinal vesicle.

		Superovulated	Control	*p* Values
Follicle isolation	Ovaries used (N)	38	18	
	Follicles cultured (N)	225	200	
	Follicle yield/ovary (Mean ± SD)	6.0 ± 0.65 *	11.1 ± 2.08	0.0151
Culture outcomes	Attachment on Day 1	N (%) 177 (78.7%) ***	N (%) 84 (42.0%)	<0.0001
	Attachment on Day 3	210 (93.3%) ***	153 (76.5%)	<0.0001
	Antrum formation	20 (8.9%)	20 (10.0%)	0.6954
	Survived till Day 12	155 (68.9%)	150 (75.0%)	0.1624
	Put into maturation	119 (52.9%)	99 (49.5%)	0.4854
Maturation outcomes	MII oocytes	76 (63.9%)	59 (59.6%)	0.5180
	MI oocytes	28 (23.5%) **	10 (10.1%)	0.0093
	GV oocytes	13 (10.9%) **	28 (28.3%)	0.0011

**Table 2 jcm-10-02757-t002:** Follicular sizes during culture (mean follicle diameter ± standard deviation, µm) of follicles from superovulated ovaries vs. controls (** *p* < 0.01, *** *p* < 0.001).

	Day 0	Day 1	Day 3	Day 5	Day 7	Day 9	Day 11	Day 12
Superovulated	116.6 ± 9.1	121.6 ± 11.0 **	131.6 ± 21.4 ***	161.5 ± 46.7 ***	211.2 ± 84.9 ***	279.0 ± 134.6	375.5 ± 157.3	460.1 ± 150.1
Control	115.3 ± 10.0	118.5 ± 10.3	122.3 ± 13.4	139.4 ± 26.1	177.1 ± 55.8	256.7 ± 119.5	374.3 ± 159.2	450.1 ± 178.6

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
