# Peer review of "Feasibility of Secondary Follicle Isolation, Culture and Achievement of In-Vitro Oocyte Maturation from Superovulated Ovaries: An Experimental Proof-of-Concept Study Using Mice"

_jcm, 2021, doi:10.3390/jcm10132757_

Round 1
Reviewer 1 Report
Improvement of treatment and survival of adolescent oncology patients brings preservation of fertility to central survivorship issue. This study investigated the feasibility of in vitro culturing secondary follicles in animal model after gonadothropins. The results provides data on the possiblity to obtain secondary follicles in women who undergo cryopreservation and failed to produce mature oocystes. The manuscript is well designed. Material and methods are described appropriately. Results are presented clrealy and discussed. The authors used appropriate references. I assess this study as interesting and valuable.
Author Response
We thank the reviewers and editor for the constructive comments that have helped us to improve our manuscript. We have also revised the language and style, as recommended.
Reviewer 2 Report
In this article, Hao et al. present an original research on the Feasibility of follicle isolation and achievement of in vitro oocyte maturation from superovulated ovaries – An experimental study using mouse.
line 38- what are the success rates for oocyte preservation and pregnancy
line 43- Cryopreservation of ovarian tissue - what are the success rates
line 141 - why Day 5, 9 and 12 were used
Table 1 and Table 2- please explain the abbreviations used (i.e. GV, MI, MII)
line 172- Table 2 is difficult to follow
line 202-203 - please improve the language used in the paragraph
line 266- please provide more limitations to the study; how could the amount of GnRH used influence the follicle growth? what might have happened if more or less GnRH was administered?
There is no discussion on how timing influenced the diameter of the follicle and the estradiol level nor when would be the perfect timing to collect the follicles
I believe the discussion should be improved by showing more pros and cones of the experiment.
Author Response
We thank the reviewers and editor for the constructive comments that have helped us to improve our manuscript. We have also revised the language and style, as recommended.
Point 1: line 38- what are the success rates for oocyte preservation and pregnancy
Response 1: Thank you for your comment. Information on the pregnancy rate of using cryopreserved mature oocytes has been added to the manuscript, line 40.
Point 2: line 43- Cryopreservation of ovarian tissue - what are the success rates
Response 2: The success rate of ovarian tissue cryopreservation according to current data was added to line 51.
Point 3: line 141 - why Day 5, 9 and 12 were used
Response 3: We have improved the text to clarify this. An explanation was added to paragraph at lines 252-256.
Point 4: Table 1 and Table 2- please explain the abbreviations used (i.e. GV, MI, MII)
Response 4: The abbreviations have been explained in the text and in the figure legend.
Point 5: line 172- Table 2 is difficult to follow
Response 5: We have reformatted the Table 2 and made improvements for the presentation of the data.
Point 6: line 202-203 - please improve the language used in the paragraph
Response 6: Thank you. We have rephrased the paragraph and we have revised the language throughout the manuscript.
Point 7: line 266- please provide more limitations to the study; how could the amount of GnRH used influence the follicle growth? what might have happened if more or less GnRH was administered?
Response 7: Thank you for your comment. We have improved the discussion section regarding the limitations, please see lines 290-296.
We did not use GnRH in follicle culture, we used recombinant FSH, at the standardized concentration previously established. We present the complete information in the methods section, according to its use in previous studies on secondary follicle culture using mice.
Point 8: There is no discussion on how timing influenced the diameter of the follicle and the estradiol level nor when would be the perfect timing to collect the follicles
Response 8: Thank you for your suggestion. We have explained in the discussion on the changes in follicle size over time and also the changes in estradiol levels. We have also improved the text to explain which follicles and at which size these were collected for maturation. Please see this in discussion (lines 252-256) and on the timing for follicle collection to IVM was added to lines 270-271.
Point 9: I believe the discussion should be improved by showing more pros and cons of the experiment.
Response 9: Thank you for your suggestion. We have developed the discussion about the pros (lines 278-287) and about the limitations (lines 290-296).
Reviewer 3 Report
I read with interest the paper by Hao et al. describing an experimental study on mice to evaluate the feasibility of IVM of immature oocytes from superovulated ovaries.
I believe that the introduction would benefit from further contestualization abour current clinical practice. Moreover, while I understand that this is an animal study, I would like to read more on the final outcome: the conclusion should states clearer how the discovery could potentially change clinical practice.
Indeed, IVM is (esperimentally) used both after transvaginal OPU from unstimulated ovaries or on immature oocytes retrieved from ovarian tissue. Ovarian tissue cryopreservation is never performed after COS because there is evidence that COS causes extensive damage to the ovarian cortex tissue (due to the expansion of ovarian cortex+the blood following aspiration of follicles). Indeed, it is more common to do an unilateral oophorectomy for ovarian tissue cryopreservation +/- OTO-IVM and then, if it is possible to postpone oncological tratment, COS on the remaining ovary. Another option is to wait for the endocrine profile to return to basal before OTC (not relevant for the current study). I believe that IVM on immature oocytes after COS could be useful to obtain few extra oocytes, when some of them are immature at OPU but it would not change dramatically the efficience of the technique.
It is not clear in the manuscript if the Authors are thinking (as possible clinical application) of IVM on immature oocytes retrieved during OPU together with mature ones to increase the final number or of retrieving laparoscopically ovarian tissue right after COS.
If the second option, I don't think it gives additional benefit.
If first option, since IVM success is still suboptimal/to be improved, it could be nice to have a chance more instead of just disposing of all immature oocytes, but the final effect on LBR is probably minimal.
Author Response
We thank the reviewers and editor for the constructive comments that have helped us to improve our manuscript. We have also revised the language and style, as recommended.
Point 1: I believe that the introduction would benefit from further contestualization about current clinical practice.
Response 1: Thank you for your suggestion. We have improved the introduction accordingly.
Point 2: Moreover, while I understand that this is an animal study, I would like to read more on the final outcome: the conclusion should states clearer how the discovery could potentially change clinical practice.
Response 2: We have followed the reviewer’s recommendation and have improved the texts in the discussion accordingly, to indicate potential clinical usefulness, lines 278-287.
Point 3: Indeed, IVM is (experimentally) used both after transvaginal OPU from unstimulated ovaries or on immature oocytes retrieved from ovarian tissue. Ovarian tissue cryopreservation is never performed after COS because there is evidence that COS causes extensive damage to the ovarian cortex tissue (due to the expansion of ovarian cortex+the blood following aspiration of follicles). Indeed, it is more common to do an unilateral oophorectomy for ovarian tissue cryopreservation +/- OTO-IVM and then, if it is possible to postpone oncological treatment, COS on the remaining ovary. Another option is to wait for the endocrine profile to return to basal before OTC (not relevant for the current study). I believe that IVM on immature oocytes after COS could be useful to obtain few extra oocytes, when some of them are immature at OPU but it would not change dramatically the efficience of the technique.
Response 3: Thank you for your comment. We agree with the reviewer. We have conducted this study as proof of concept to investigate the feasibility of obtaining mature oocytes from secondary follicles retrieved from ovaries previously superovulated. The further development of the method is absolutely needed to improve the final outcomes. We believe that if this approach would be further developed as a clinical method, the efficiency should be investigated, as it could benefit the patients in special situations, for example after a failed attempt to recover oocytes for cryopreservation.
Point 4: It is not clear in the manuscript if the Authors are thinking (as possible clinical application) of IVM on immature oocytes retrieved during OPU together with mature ones to increase the final number or of retrieving laparoscopically ovarian tissue right after COS.
If the second option, I don't think it gives additional benefit.
If first option, since IVM success is still suboptimal/to be improved, it could be nice to have a chance more instead of just disposing of all immature oocytes, but the final effect on LBR is probably minimal.
Response 4: Thank you for your comment. We have revised the manuscript sections and also modified the title to make more clear for the reader about the objective of our study. The clinical situation and potential indication for the method described in our manuscript was added to lines 67-71 in introduction and lines 278-287 in discussion. Till our knowledge, patients that receive COS do not proceed currently to performing an unilateral oophorectomy for ovarian tissue cryopreservation, thus if COS fails and the initiation of cancer treatment could not be delayed, no other options could be provided. In this context, as a rescue method, the ovaries could be retrieved to perform secondary follicle isolation and culture to obtain mature oocytes to be cryopreserved. We have tried to explain this better in the manuscript’s introduction and discussion parts. Additionally, ovarian tissue cryopreservation would probably need to be adapted to work with an ovary that has been superovulated, and the success rate may be lower than for unstimulated ovaries. As far as we know that study has not been performed, so this is just a guess. If secondary follicles could also be retrieved in this hypothetical situation, these could represent an additional resource of mature oocytes to be cryopreserved, to increase the success rate of conducting fertility preservation.